# Seamless trials in oncology: A cross-sectional analysis of characteristics and reporting

**Katarzyna Klas**[1,2], **Karolina Strzebonska**[1], **Paola Buedo**[1], **Alicja Włodarczyk**[1], **Samuel Gordon**[1], **Paulina Kaszuba**[1], **Maciej Polak**[1,3], **Marcin Waligora**[1] *

**1** Faculty of Health Sciences, Research Ethics in Medicine Study Group (REMEDY), Jagiellonian University Medical College, Krakow, Poland, **2** Doctoral School of Medical and Health Sciences, Jagiellonian University Medical College, Krakow, Poland, **3** Faculty of Health Sciences, Department of Epidemiology and Population Studies, Institute of Public Health, Jagiellonian University Medical College, Krakow, Poland

* m.waligora@uj.edu.pl

**Data Availability Statement:** Data used for our analysis are publicly available on the Open Science Framework website (https://osf.io/m346x/).

**Funding:** This study was funded by the National Science Center, Poland, UMO-2021/41/B/HS1/

## Abstract

### Objectives

Seamless clinical trials have received much attention as a possible way to expedite drug development. The growing importance of seamless design can be seen in oncology research, especially in the early stages of drug development. Our objective is to examine the basic characteristics of seamless early-phase oncology trials registered on the Clinical-Trials.gov database and to determine their results reporting rates. We also aim to identify factors associated with results reporting.

### Methods

Cross-sectional study. We defined seamless early-phase trials as either those registered as Phase 1/2 or Phase 1 with planned expansion cohort(s). Using the ClinicalTrials.gov registry, we searched for interventional cancer clinical trials with primary completion date (PCD) between 2016 and 2020. After trial selection, we performed manual data extraction based on the trial record description and the results posted in the trial registry. We used logistic regression to search for predictors of results reporting. Protocol: https://osf.io/m346x/.

### Results

We included 1051 seamless early-phase oncology trials reported as completed (PCD) between 2016 and 2020. We provided descriptive statistics including the number of patients enrolled, study start date, primary completion date, funding, type of intervention, cancer type, design details, type of endpoints, recruitment regions, and number of trial sites. Overall, only 34.7% trials reported results on ClinicalTrials.gov. The results reporting rates for 24 months was 24.0%. The overall reporting rate for Phase 1/2 studies was over three times higher than for seamless Phase 1.

### Conclusions

Our study provides cross-sectional data on seamless early-phase oncology trials registered on ClinicalTrials.gov. We highlight the challenges of the evolving clinical trial design

01123 (www.ncn.gov.pl). Authors received the funding: MW, KK, KS, MP. The funder had no role in study design, data collection and analysis, decision to publish, or preparation of the manuscript. The authors received funding for the Open Access fee from the Strategic Programme Excellence Initiative of the Jagiellonian University.

**Competing interests:** The authors have declared that no competing interests exist.

landscape and the problem of missing results in the seamless design context, which raises serious ethical concerns. Efforts should be made to adapt the functionality of the Clinical-Trials.gov database to emerging clinical trial models.

## Introduction

In recent years, the landscape for conducting clinical trials has evolved. New clinical trial design solutions are emerging with promise of greater flexibility and speed in the drug development process [1,2]. Seamless clinical trials, a type of adaptive clinical trial design, are one such solution [3].

Seamless trials often combine two distinct phases of drug development, allowing for the concurrent assessment of objectives traditionally assessed in separate trials [1]. In the traditional approach, the clinical trials of a drug are conducted sequentially in separate trials from Phase 1 to Phase 3 prior to marketing approval. Phase 1 trials focus primarily on evaluating initial safety and dose tolerance. Phase 2 trials aim to analyze preliminary efficacy. Phase 3 trials evaluate the investigational drug's comparative benefits. Seamless designs may adopt a Phase 1/2 trial format, concurrently addressing safety and efficacy as one trial. Similarly, a Phase 2 and a Phase 3 trial may be combined into a seamless Phase 2/3 trial. By merging phases and moving seamlessly from one phase to another, the gap between initiating a separate phase (as in the traditional mode) is eliminated [3–5].

Seamless clinical trials may also refer to Phase 1 trials intending to include an expansion cohort(s) after completion of the dose escalation component [6–9]. Additional cohorts of patients may be seamlessly added to the Phase 1 trial to further evaluate the drug within the same trial. As a result, a Phase 1 study with expansion cohorts typically enrolls a larger number of participants. It is not primarily limited to analyzing the safety and determining the tolerated dose of the therapy being tested [10]. It also may assess other aspects typically associated with Phase 2 trials, such as preliminary efficacy [8].

Clinical trials using the "seamless design" methodology are used across a range of medical disciplines, including for example cardiology, pulmonology, vaccine development [1,11,12]. Seamless design was also used in clinical trials conducted during the global pandemic caused by the SARS-CoV-2 virus [13,14]. Nevertheless, the increasing significance of seamless design is particularly evident in the context of oncology research [4,5,15,16]. Advances in oncology drug development, including precision medicine approaches and novel targeted and immuno-therapies, have stimulated the need to detect antitumor activity as early as possible [7]. The increased use of seamless design in the early stages of new anticancer drug development is particularly striking. The purpose is to enable the selection of the most promising therapies at an early stage, which eventually may result in earlier access to treatment for cancer patients [3,4,8,15,17].

Our objective is to examine the basic characteristics of seamless early-phase oncology trials that are registered on the ClinicalTrials.gov database. We aim to determine their results reporting rates and identify factors associated with faster results reporting.

## Methods

This cross-sectional meta-research study follows a pre-specified protocol available on the Open Science Framework (OSF) website (https://osf.io/m346x/) [18]. The study relies on publicly accessible data and follows the Strengthening the Reporting of Observational studies in

Epidemiology (STROBE) reporting guideline for cross-sectional studies (see S1 Table) [19]. An overview of the methods is presented in Fig 1.

## Sample and trial selection

We identified our sample on ClinicalTrials.gov. We chose this platform because it is a well-known clinical trial registry and a repository of clinical trial data from around the world. It is also a relational database that is used to conduct scientific research on the design of clinical trials [20]. We defined seamless early-phase trials as either those registered as Phase 1/2 or Phase 1 with planned expansion cohort(s). Using the advanced search function available on the ClinicalTrials.gov registry, we searched for interventional cancer clinical trials with primary completion date (PCD) between 2016 and 2020 that were registered as Phase 1 and Phase 1/2 trials. We chose the time range for two reasons: to obtain a sample of reasonable and feasible size, and to give each trial at least two years after the primary completion date to submit summary results to the ClinicalTrials.gov database. We automatically downloaded the dataset as a CSV file on 01/03/2023. We used the following search string to generate the sample: Completed Studies | Interventional Studies | Cancer | Phase 1, 2 | Primary completion from 01/01/2016 to 12/31/2020. See more details about search strategy in S2 Table.

During trial selection, we automatically excluded trials registered as Phase 2 and Phase 2/3 from our dataset. Then, we performed manual assessment of trial eligibility, which required evaluation of the full trial record on the ClinicalTrials.gov registry. We searched for trials testing drugs and biologics for antitumor activity. Both adult and pediatric studies evaluating both solid tumors and hematologic malignancies were included. We conducted additional assessments for eligible Phase 1 trials. Our aim was to specifically include Phase 1 trials meeting the criteria for a seamless design. To ensure this, we examined whether the descriptions of these trials mentioned the expansion cohort directly. In the absence of direct information on the expansion cohort, we checked the study description for information on an additional group of patients recruited into the study after the end of the dose-escalation component. If neither criterion was met, we excluded the Phase 1 study from further analysis. The full inclusion and exclusion criteria can be found in the protocol [18].

Prior to manual assessment, each reviewer received training and conducted a pilot. During the manual trial selection process, each trial record was assessed independently by two reviewers (KK assessed all trial records, and the second reviewer was KS, PB, or AW). Disagreements were resolved by discussion and, if necessary, by a referee (MW).

## Data collection

We created and piloted an extraction form [18] (see S1 File). For each of the included trials, we performed a double-checked manual extraction of the data. Disagreements were reconciled through discussion and, if necessary, the involvement of an arbiter. We extracted some of the data directly from the dataset downloaded from ClinicalTrials.gov. This included the start and end dates of the study, the age of the study participants, the number of enrolled patients, the type of funder, whether the results were reported, and, if so, when they were first reported.

We assessed other variables using data from the ClinicalTrials.gov website (online access). Based on the record description, we determined the design details (randomization, masking, interventional model), the type of intervention, the number of drugs evaluated, the type of cancer, and the number and location of study sites. We also confirmed whether the trial description reported the trial phase consistently. This involved searching for any discrepancies in phase reporting across different sections of the study record on ClinicalTrials.gov. An example of a discrepancy was a trial registered as Phase 1/2, yet only Phase 1 was mentioned in the

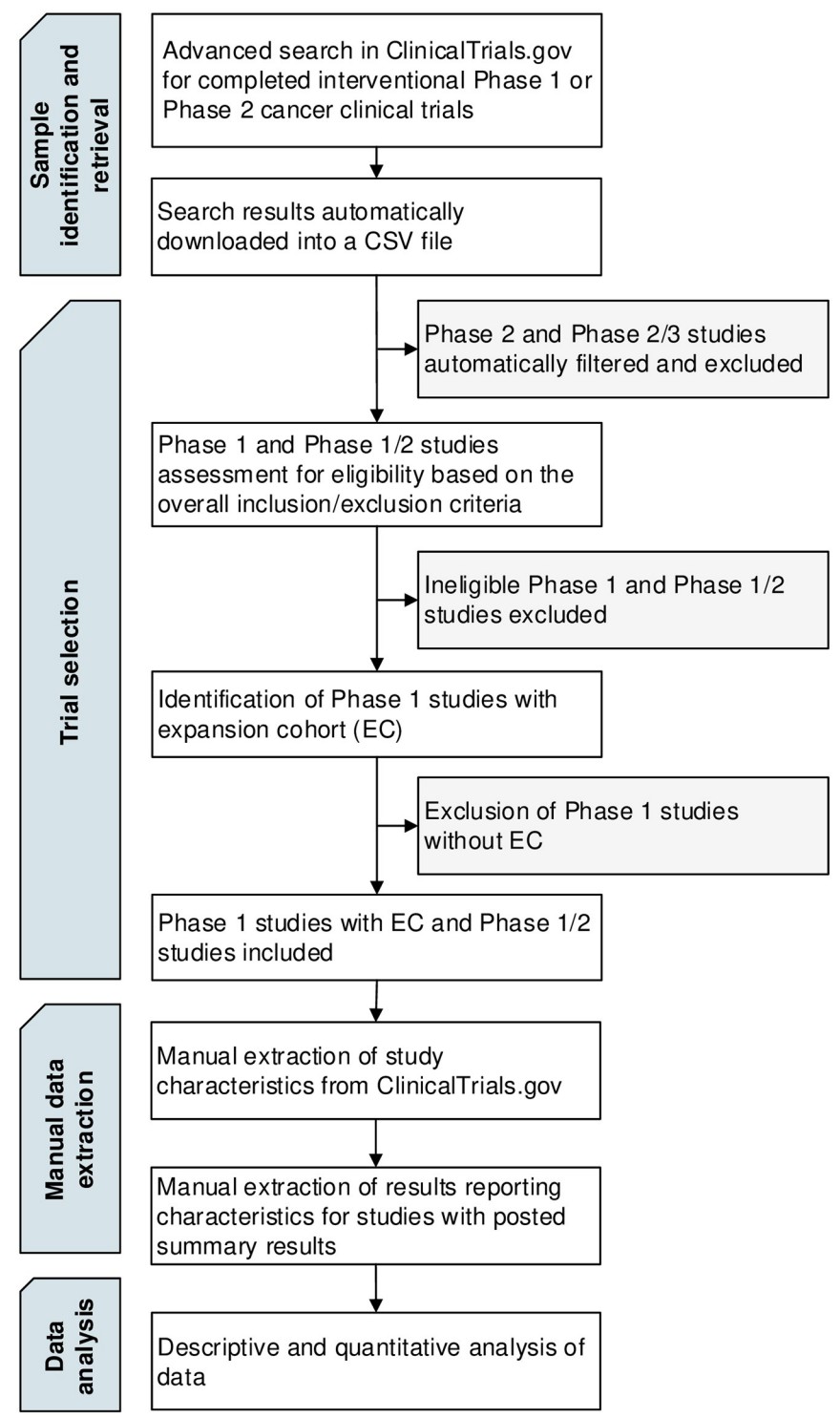

**Fig 1. Method overview.**

record description. We also examined whether the study record reported the presence of each seamless trial stage. In this analysis, "trial stage" refers to Phase 1 or Phase 2 for Phase 1/2 trials, and to the stage preceding the enrollment of expansion cohort(s) and the expansion cohort(s) themselves for seamless Phase 1 trials.

In addition, we verified whether included Phase 1/2 trials met the definition of seamless Phase 1 trials. For those trials that directly mentioned "expansion cohort", we checked where this phrase was mentioned in the study record within the specific sections on ClinicalTrials. gov. This was a sequential process. First, we checked the Title section, then the other elements of the Study Overview section (i.e. Brief and Detailed Description), the Arms and Interventions, and the Participation Criteria.

For all studies in our sample, we assessed the types of endpoints. We checked the number of studies that reported the following endpoints: dose-limiting toxicities (DLTs)/maximum tolerated dose (MTD), recommended phase 2 dose (RP2D), treatment-related adverse events (TRAEs), progression-free survival (PFS), overall survival (OS), response rates (RR) (e.g. overall response rate) and pharmacokinetic measures. Synonymous terms referring to these endpoints were also considered. We checked if the primary endpoints were separately reported for each seamless trial stage.

For studies with reported results, we assessed the characteristics of the results reporting. Results on ClinicalTrials.gov are usually presented as a separate table for participant characteristics and outcome measures. Our goal was to explore whether the data for each trial stage were reported separately or together. To classify a Phase 1/2 trial as one with separately reported data for each trial stage, results should be reported separately for Phase 1 and Phase 2. Likewise, for a seamless Phase 1 study, results should be reported separately for the stage preceding the initiation of the expansion cohort(s) and for the expansion cohort stage.

As ClinicalTrials.gov is a continuously updated database, the reference point for data extraction was the version of the record in effect at the time of the data search and download. If the date of the last record update was after 01/03/2023, we extracted data from the historical version of the trial record. We present the details about trial characteristics categorization in S3 Table.

## Statistical analysis and tools

We used Cohen's kappa to calculate the inter-rater agreement between the reviewers for the study selection. We presented the descriptive statistics for the both the entire sample of seamless early-phase oncology trials and separately for each phase. We reported counts and percentages for categorical variables. We assessed trial duration as time from study start date to primary completion date. We calculated the time to results reporting as the time from the primary completion date to the date when the results were first reported. We described the time to results reporting by median with interquartile range (IQR). We reported on rates of trials that posted the results on ClinicalTrials.gov (overall and within 24 months since primary completion date).

We used the chi-square test or Exact Fisher test to examine the differences in results reporting between trials characteristics. To compare time to reporting results between trial characteristics, we used Mann-Whitney U test or Kruskal-Wallis test. We performed the multivariable logistic regression analysis to find the independent predictors of reporting results within two years. As we considered clinical trials with a primary completion date between 2016–2020, we evaluated the predictors with respect to two years (24 months), so that each trial had at least two years to report results. We chose 24 months for our analysis with reference to the World Health Organization's "Joint statement on public disclosure of results from clinical trials" [21].

This statement indicates that the primary results of a clinical trial should be reported in a registry within 12 months and published in a peer-reviewed journal within 12 months. Given the expected limited number of clinical trials that had reported results within the 12-month window, we decided to perform our statistical analysis in the 24-month window. All of the characteristics that were statistically significant in the univariable analysis were included in the multivariable model. Then backward elimination method was applied (P<0.1). We presented the results of regression model as odds ratio (OR) with 95% confidence interval (CI). We examined time to results reporting using Kaplan-Meier curves and compared using the log-rank test. For the study not reporting results, we censored the timeline on March 1, 2023. We performed analysis using Microsoft Excel and IBM SPSS Statistics for Windows, Version 28.0. (2021) Armonk, NY: IBM Corp. For statistical inference, N/R (not reported) was treated as missing data. All tests were 2-sided, and P-value less than 0.05 was considered statistically significant.

## Results

### Trial characteristics

We included 1051 early-phase seamless oncology trials reported as completed (PCD) between 2016 and 2020, including 562 (53.5%) seamless Phase 1 and 489 (46.5%) trials registered as Phase 1/2 (see S1 Fig for flow diagram). The inter-rater agreement (Cohen's kappa) between the reviewers during the trial selection was 0.78 (95% CI: 0.75–0.80). The characteristics of all included trials are presented in Table 1 and S4 Table, which present the characteristics of seamless Phase 1 and Phase 1/2 trials separately). The median number of enrolled patients was 45 (IQR: 25–85). Median trial duration was 3.8 (IQR: 2.7–5.3) years. Of the 1051 studies, 813 (77.4%) were at least partially industry funded, 987 (93.9%) were adult studies, 922 (87.7%) were non-randomized and 733 (69.7%) were multi-site. Almost half of the trials (512, 48.7%) were conducted in North America and 476 (45.3%) were conducted exclusively in the United States.

Most trials evaluated multiple agents (654, 62.2%). We found that 238 (22.6%) trials evaluated immunotherapies, 263 (25.0%) targeted therapies, 67 (6.4%) cytotoxic therapies. Other types of therapy were reported in 25 (2.4%) trials and 458 (43.6%) trials evaluated combinations of at least two of the above. The majority of trials involved solid cancers (752, 71.6%). Over half (530, 50.4%) evaluated more than one type of cancer. The most common types of cancer mentioned in study descriptions on ClinicalTrials.gov include lung cancer, breast cancer, lymphoma, leukemia, ovarian cancer, and colorectal cancer. These indications were mentioned in at least 10% of the assessed trials. See S5 Table for more details.

We found that the description of the study was consistent with phase registration in the case of 1010 (96.1%) trials. The study record reported the presence of each seamless trial stage in the case of 903 (85.9%) trials. Of the 489 of Phase 1/2 trials, 301 (61.6%) met the criteria for Phase 1 with expansion cohort(s). The expansion cohort was directly mentioned in 665/1051 (63.3%) trials (i.e. 518/562 (92.2%) for Phase 1 and 147/489 (30.1%) for Phase 1/2). Of these, only 92/665 (13.8%) mentioned expansion cohort(s) in the "Title" and more than half (376/665, 56.5%) in "Brief and Detailed Description". See S6 Table for more details.

Of the endpoints evaluated, MTDs or DLTs occurred in 747/1051 (71.1%) studies, and RP2D in 168 (16.0%). PFS, OS and RR were reported in 518 (49.3%), 388 (36.9%) and 856 (81.4%) trials, respectively. TRAEs occurred in 142 (13.5%) studies. Pharmacokinetic measures were identified in 559 (53.2%) studies. Only 189 (18.0%) studies reported primary endpoints by seamless trial stage. We found that MTD or DLTs, RP2D, and TRAEs were mainly reported as primary endpoints, while OS, PFS, RR, and pharmacokinetic measures were reported as

**Table 1. Characteristics of oncology seamless early-phase clinical trials completed between 2016 and 2020 registered on ClinicalTrials.gov.**

| Characteristic[a] | N (%) |
| --- | --- |
| **Number of trials** | 1051 (100%) |
| **Phase** | |
| Phase 1 | 562 (53.5%) |
| Phase 1/2 | 489 (46.5%) |
| **Enrolled participants** | |
| 1–50 | 594 (56.5%) |
| 51–100 | 246 (23.4%) |
| 101–150 | 105 (10.0%) |
| 151–200 | 48 (4.6%) |
| >200 | 58 (5.5%) |
| **Study start date** | |
| 2013–2014 | 325 (30.9%) |
| 2015–2016 | 301 (28.6%) |
| 2011–2012 | 188 (17.9%) |
| 2017–2018 | 138 (13.1%) |
| ≤2010 | 91 (8.7%) |
| 2019–2020 | 8 (0.8%) |
| **Primary completion date** | |
| 2019 | 239 (22.7%) |
| 2020 | 224 (21.3%) |
| 2017 | 205 (19.5%) |
| 2018 | 193 (18.4%) |
| 2016 | 190 (18.1%) |
| **Funder type** | |
| Industry | 596 (56.7%) |
| Non-industry | 238 (22.6%) |
| Partially-industry | 217 (20.6%) |
| **Study population age[b]** | |
| Adults | 987 (93.9%) |
| Both | 61 (5.8%) |
| Pediatric | 3 (0.3%) |
| **Type of intervention** | |
| Mixed[c] | 458 (43.6%) |
| Targeted therapy | 263 (25.0%) |
| Immunotherapy | 238 (22.6%) |
| Cytotoxic therapy | 67 (6.4%) |
| Other | 25 (2.4%) |
| **Number of drugs evaluated in the study** | |
| Multiple agents | 654 (62.2%) |
| Single agent | 397 (37.8%) |
| **Type of cancer** | |
| Solid | 752 (71.6%) |
| Hematological | 250 (23.8%) |
| Both | 46 (4.4%) |
| Not reported | 3 (0.3%) |
| **Number of cancer types** | |

*(Continued)*

**Table 1.** (Continued)

| Characteristic[a] | N (%) |
|---|---|
| Multiple | 530 (50.4%) |
| Single | 521 (49.6%) |
| **Masking** | |
| Open label | 1032 (98.2%) |
| At least double-blind | 15 (1.4%) |
| Single blind | 1 (0.1%) |
| Not reported | 3 (0.3%) |
| **Randomization** | |
| Non-randomized | 922 (87.7%) |
| Partially randomized | 56 (5.3%) |
| Randomized | 55 (5.2%) |
| Not reported | 18 (1.7%) |
| **Interventional model** | |
| Single group assignment | 622 (59.2%) |
| Parallel assignment | 268 (25.5%) |
| Sequential assignment | 144 (13.7%) |
| Factorial assignment | 5 (0.5%) |
| Cross-over assignment | 3 (0.3%) |
| Not reported | 9 (0.9%) |
| **Number of trial's sites** | |
| Multi-site | 733 (69.7%) |
| Single-site | 307 (29.2%) |
| Not reported | 11 (1.0%) |
| **Recruitment regions #1** | |
| North America | 512 (48.7%) |
| Mixed | 254 (24.2%) |
| Europe | 147 (14.0%) |
| Asia | 112 (10.7%) |
| Australia | 14 (1.3%) |
| South America | 1 (0.1%) |
| Not reported | 11 (1.0%) |
| **Recruitment regions #2** | |
| United States (US) | 476 (45.3%) |
| Non-US | 298 (28.4%) |
| Multicenter including US | 266 (25.3%) |
| Not reported | 11 (1.0%) |

[a]The characteristics are presented in descending order of frequency, with the exception of the "not reported" category.

[b]The categorization was adopted directly from the ClinicalTrials.gov registry. We chose "Pediatrics" for the "Child" category or age under 18 (if the specific age range was reported). And "Adult" for the "Adult" and "Older Adult" categories and age over 18.

[c]The "Mixed" for "type of intervention" category encompasses combinations of at least two drugs from two distinct categories, including targeted therapy, immunotherapy, cytotoxic therapy, and other treatments.

US: United States.

secondary endpoints. See S7 Table for details on the division into primary and secondary endpoints.

### Rates of results reporting

With a median follow-up of 518 (IQR: 412–801) days, 365 of 1051 trials (34.7%) reported results on ClinicalTrials.gov. Two trials (0.5%) published results before their primary completion date. Overall, more than half (269/489, 55.0%) of the Phase 1/2 trials posted results on ClinicalTrials.gov, whereas less than one in five Phase 1 trials (96/562, 17.1%) did so. The overall reporting rate for Phase 1/2 studies was therefore more than three times higher than for seamless Phase 1 studies (see Table 2 and S2 Fig). The rate of results reported within 24 months was 24.0% for all trials, 10.7% for Phase 1 and 39.3% for Phase 1/2 (see Table 2). We present the time to results reporting for all trials in our sample in Fig 2 and separately for each phase in S2 Fig. For comparison of time to reporting results based on trial characteristics see S8 Table.

### Characteristics of the results reports

We found that patient characteristics and outcome results were reported separately for each trial stage in 176/365 (48.2%) trials, reported together (not broken down by trial stage) in 145 trials (39.7%), and reported only for the first stage in 17 (4.7%) trials. In 27 (7.4%) trials, the results reporting was inconsistent for participants and outcomes (e.g. characteristics of participants were reported separately for each stage but together for the outcomes). Among the clinical trials with results separately reported, the median number of expansion cohort(s) or Phase 2 cohort(s) was 2 (IQR: 1–3). The median number of patients enrolled in all expansion/Phase 2 cohorts was 45 (IQR: 24–93).

### Predictors of results reporting

We present the comparison of trials with results reported within 24 months in Table 2. A higher proportion of results reported within 24 months was associated with Phase 1/2 trials, US-led trials, trials with a later primary completion date, partially randomized trials, trials with larger recruitment targets, and trials evaluating multiple agents, hematologic cancers, and mixed population age.

In multivariable analysis (Table 3), we found five independent predictors of reporting results within 24 months. The odds of reporting (OR) were more than eight times higher for Phase 1/2 trials than for Phase 1 trials. We found that compared to solid tumors, hematologic malignancies had 1.5 times higher odds of reporting. Trials conducted in or involving the United States had 6.7 and 8 times higher odds of reporting, respectively, than non-US trials. The odds of reporting results were 1.58 times higher for trials recruiting more than 50 and fewer than 100 participants compared to trials recruiting 50 or fewer participants. The later the primary completion date of the trial, the greater the odds of results being reported on ClinicalTrials.gov (OR = 1.4).

### Discussion

Seamless design is relatively new in clinical trials. This clinical trial model has gained popularity as a way to accelerate drug development. The interest in seamless design in oncology was sparked by the successful Phase 1 trials with expansion cohorts that led to accelerated approval of pembrolizumab in melanoma in less than 4 years in 2014 [4,5,15]. Currently, early phase dose-expansion cohort studies are an important type of study to support the Food and Drug Administration approval of targeted anticancer drugs [22].

**Table 2. A comparative analysis of results reporting rates based on the characteristics of the trials.**

| Characteristic | Trials with results reported within 24 months | | Trials with results reported (overall) | |
|---|---|---|---|---|
| | N (%)[a] | P-value | N (%)[a] | P-value |
| **All trials** | 252 (24.0%) | Not applicable | 365 (34.7) | Not applicable |
| **Phase** | | | | |
| Phase 1 | 60 (10.7%) | <0.001 | 96 (17.1%) | <0.001 |
| Phase 1/2 | 192 (39.3%) | | 269 (55.0%) | |
| **Enrolled participants** | | | | |
| 1–50 | 111 (18.7%) | <0.001 | 166 (27.9%) | <0.001 |
| 51–100 | 66 (26.8%) | | 95 (38.6%) | |
| 101–150 | 38 (36.2%) | | 52 (49.5%) | |
| 151–200 | 13 (27.1%) | | 18 (37.5%) | |
| >200 | 24 (41.4%) | | 34 (58.6%) | |
| **Study start date** | | | | |
| ≤2010 | 28 (30.8%) | 0.036 | 41 (45.1%) | 0.006 |
| 2011–2012 | 43 (22.9%) | | 74 (39.4%) | |
| 2013–2014 | 68 (20.9%) | | 111 (34.2%) | |
| 2015–2016 | 86 (28.6%) | | 106 (35.2%) | |
| 2017–2018 | 24 (17.4%) | | 30 (21.7%) | |
| 2019–2020 | 3 (37.5%) | | 3 (37.5%) | |
| **Primary completion date** | | | | |
| 2016 | 31 (16.3%) | <0.001 | 61 (32.1%) | 0.706 |
| 2017 | 35 (17.1%) | | 71 (34.6%) | |
| 2018 | 46 (23.8%) | | 71 (36.8%) | |
| 2019 | 60 (25.1%) | | 78 (32.6%) | |
| 2020 | 80 (35.7%) | | 84 (37.5%) | |
| **Funder type** | | | | |
| Industry | 135 (22.7%) | 0.458 | 200 (33.6%) | 0.623 |
| Non-industry | 59 (24.8%) | | 88 (37.0%) | |
| Partially-industry | 58 (26.7%) | | 77 (35.5%) | |
| **Study population age** | | | | |
| Adults | 230 (23.3%) | 0.006 | 334 (33.8%) | 0.009 |
| Pediatric | 3 (100.0%) | | 3 (100.0%) | |
| Both | 19 (31.1%) | | 28 (45.9%) | |
| **Number of drugs evaluated in the study** | | | | |
| Single agent | 69 (17.4%) | <0.001 | 101 (25.4%) | <0.001 |
| Multiple agents | 183 (28.0%) | | 264 (40.4%) | |
| **Type of cancer** | | | | |
| Solid | 167 (22.2%) | <0.001 | 252 (33.5%) | 0.006 |
| Hematological | 80 (32.0%) | | 104 (41.6%) | |
| Both | 5 (10.9%) | | 9 (19.6%) | |
| **Number of cancer types** | | | | |
| Single | 136 (26.1%) | 0.109 | 207 (39.7%) | <0.001 |
| Multiple | 116 (21.9%) | | 158 (29.8%) | |
| **Masking** | | | | |
| Open label | 249 (24.1%) | 1 | 354 (34.3%) | 1 |
| Single blind | 0 (0.0%) | | 1 (100.0%) | |
| At least double-blind | 3 (20.0%) | | 10 (66.7%) | |
| **Randomization** | | | | |

*(Continued)*

**Table 2.** (Continued)

| Characteristic | Trials with results reported within 24 months | | Trials with results reported (overall) | |
|---|---|---|---|---|
| | N (%)[a] | P-value | N (%)[a] | P-value |
| Non-randomized | 212 (23.0%) | 0.018 | 302 (32.8%) | <0.001 |
| Partially randomized | 22 (39.3%) | | 38 (67.9%) | |
| Randomized | 11 (20.0%) | | 14 (25.5%) | |
| **Number of trial's sites** | | | | |
| Single-site | 64 (20.8%) | 0.182 | 92 (30.0%) | 0.05 |
| Multi-site | 181 (24.7%) | | 266 (36.3%) | |
| **Location** | | | | |
| United States (US) | 138 (29.0%) | <0.001 | 199 (41.8%) | <0.001 |
| Multicenter including US | 82 (30.8%) | | 115 (43.2%) | |
| Non-US | 25 (8.4%) | | 44 (14.8%) | |

[a]Percentages are calculated as the ratio of clinical trials with results to all trials within each trial characteristic category. For example, the reported percentage of 10.7% for Phase 1 trials in the "Trials with results reported within 24 months" column refers to the ratio of the number of Phase 1 trials with results reported within 24 months (60 trials) to the number of all Phase 1 trials (562 trials) in the cohort.

We used the chi-square or exact Fisher test to examine differences in results reporting by trial characteristics. P-value less than 0.05 was considered statistically significant.

The table does not include values for "not reported". For statistical inference, the variable "not reported" was treated as missing data.

US: United States.

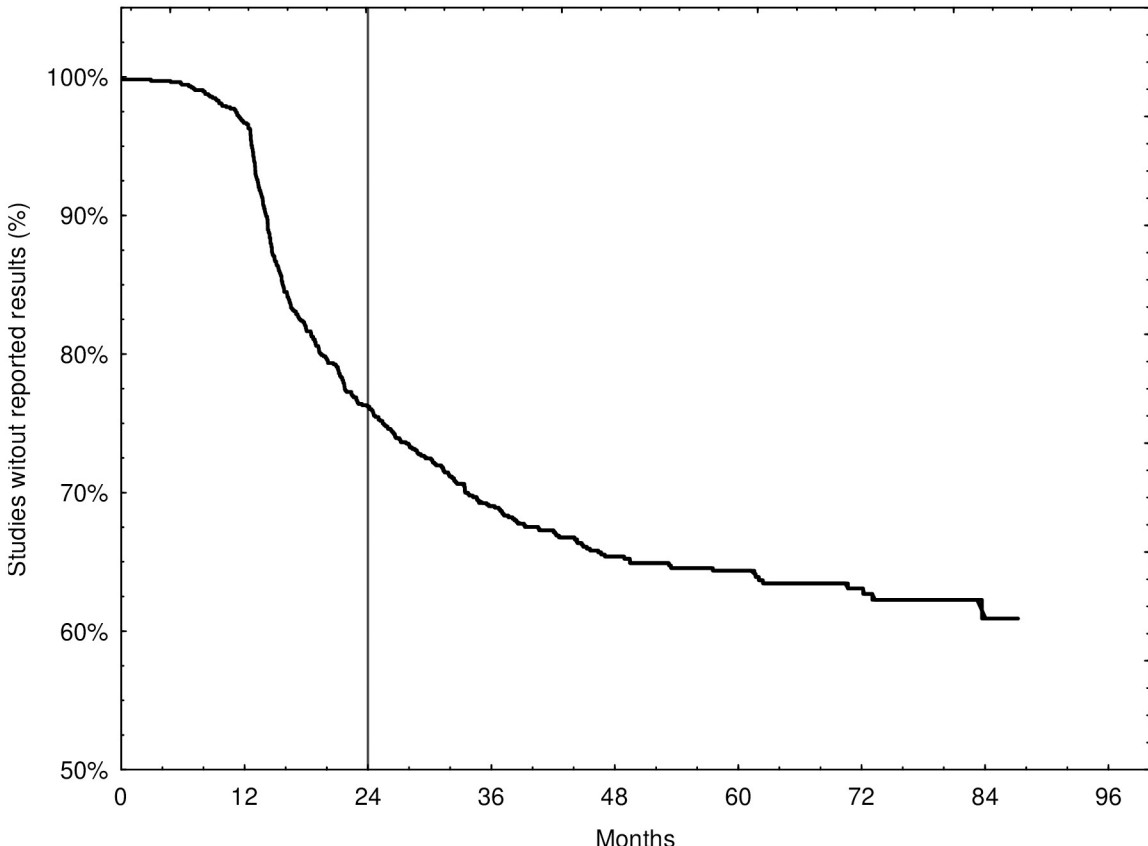

**Fig 2. Kaplan-Meier curve for the cumulative probability of not reporting clinical trial results over time.**

**Table 3. Independent predictors of results reporting within 24 months—Multivariable analysis.**

| Variable | OR | 95% CI | P-value |
|---|---|---|---|
| **Phase** | | | |
| Phase 1 | Ref | | |
| Phase 1/2 | 8.361 | (5.751–12.155) | <0.001 |
| **Type of cancer** | | | |
| Solid | Ref | | |
| Hematological | 1.506 | (1.035–2.190) | 0.032 |
| Both | 0.396 | (0.138–1.139) | 0.086 |
| **Location** | | | |
| Non-US | Ref | | |
| US | 6.731 | 4.124–10.985) | <0.001 |
| Multicenter including US | 8.051 | (4.552–14.240) | <0.001 |
| **Enrolled participants** | | | |
| 1–50 | Ref | | |
| 51–100 | 1.581 | (1.044–2.394) | 0.031 |
| 101–150 | 1.713 | (0.976–3.007) | 0.061 |
| 151–200 | 1.851 | (0.833–4.114) | 0.131 |
| >200 | 1.534 | (0.750–3.141) | 0.242 |
| **Primary completion date (per year)** | 1.403 | (1.239–1.589) | <0.001 |

In the multivariable model, the characteristics that were statistically significant in the univariable analysis were included (see Table 2, analysis for trials with results reported within 24 months) and the backward elimination method was applied (P < 0.1). US: United States. OR: odds ratio. CI: confidence interval.

In our study, we provide a comprehensive evaluation of seamless early phase clinical trials in oncology based on ClinicalTrials.gov registry data. We found that among 1051 trials only around 35% reported results on ClinicalTrials.gov. We showed that more than half of the Phase 1/2 trials disseminated the results. In contrast, less than 20% of the seamless Phase 1 trials presented results. In comparison, another study that evaluated the reporting of results among all interventional oncology trials registered on ClinicalTrials.gov from 2007 through 2017 indicates that the results were posted on ClinicalTrials.gov for 22.9% (95% CI, 22.2%-23.7%) of the trials [23]. Our findings for seamless early-phase clinical trials in oncology are consistent with other reports and confirm the need for overall improvement in results reporting, particularly with regard to posting results to clinical trials registries [23–27]. While the issues described with reporting results on the ClinicalTrials.gov registry are not new, there are a number of points to be made about the data we collected.

Seamless design promises accelerated drug development but also raises ethical challenges. As registration and dissemination of Phase 1 trial results on ClinicalTrials.gov are optional, seamless early-phase trials tend to have lower transparency [5,28]. The low rate of results reporting from seamless trials revealed by our study heightens this concern. Inadequate reporting slows the generation of knowledge about tested substances, diminishing the potential of seamless trials for acceleration. This issue extends to other innovative clinical trial designs, such as basket and umbrella trials [29–32].

With the increasing use of adaptive clinical trial methods, improvements in the functionality of clinical trial registries are needed. Clinical trial registries, such as ClinicalTrials.gov, should be optimized to enable easy tracking of the entire trial process and to accommodate the complexity of novel trial designs [5,33]. Recently, a call for improvements in the registration

and reporting of results for master protocols was published [33]. A master protocol is a clinical trial design that allows the evaluation of multiple interventional hypotheses. It typically involves multiple sub-studies. This challenges established registry tools, as they only allow presentation of condensed details about trial characteristics [1,33]. Hence, the authors call for separate registration of each sub-study for greater transparency and accountability [33].

Other authors have also commented on specific challenges in reporting seamless clinical trials [4,5]. One of the highlighted problems is that registration records on ClinicalTrials.gov may not delineate the distinct cohorts of seamless trials [5]. Therefore, trial results may be available for the entire trial rather than specific seamless trial stages. This causes difficulty in accurately tracing the trial process. In our study, we provide evidence to support this hypothesis. We found that results per seamless trial stage were presented in only half of the trials with published results. In addition, although most trials reported on the presence of each seamless trial stage, finding specific details about them proved challenging. For example, we found that less than 20% of trials reported primary outcomes specifically for each trial stage. For trials that evaluated more than one therapy, it was unclear whether this applied to the entire trial or just part of it. It was also unclear how participants would be recruited, or whether they would participate in the entire study or only in a selected stage.

Moreover, improvements should be made to help distinguish traditional Phase 1 studies from Phase 1 studies that meet seamless design criteria. We propose that for each Phase 1 study it should be clearly stated whether it will include expansion cohorts. Such an additional field could appear in the study description. In addition, the trial description should reflect the type of seamless trial being conducted. There are two types of seamless designs: inferential and operational [1]. In an operationally seamless design, the time between two phases is eliminated, but the analysis of results is performed for separate stages. In contrast, an inferential seamless design combines data from both stages in the final analysis. Actions should be taken to improve the comprehensibility of the study and the clarity of the description in this regard.

## Limitations

We have attempted to identify all eligible seamless clinical trials in our cohort. However, we acknowledge that our search strategy has some limitations. We used the term "cancer" to identify cancer clinical trials. We admit that this may have limited our sample with trials that used other terms e.g. neoplasm. Additionally, the ClinicalTrials.gov database underwent an interface change during the course of our study that affected both the description of certain elements in the clinical trial records and the functionality of the advanced clinical trial search tools. Moreover, there is a risk of some misclassification. The project relied on data reported on ClinicalTrials.gov, which varies widely in availability and accuracy. This data may be incomplete and may not have been updated since initial registration. Additionally, the evaluation of Phase 1 trials was hindered by the absence of clear information in the database indicating that the trial would follow a seamless design. Therefore, we applied a generous criterion by checking whether the trial description included information that there would be an expansion cohort or an additional group of patients after completion of the dose escalation stage. We also included clinical trials regardless of the planned number of enrolled participants. Our criterion differs from that used by other researchers [4]. They included in their analysis those seamless trials that had enrolled at least 100 participants. A significant number of the trials included in our analysis were conducted in North America. This regional concentration may limit the generalizability of the findings. Furthermore, we limited our search for clinical trial results to those reported on ClinicalTrials.gov. Results may have been reported elsewhere, potentially biasing the numbers. Further analyses should be conducted to assess the availability of results in other sources,

including journal publications or other clinical trial registries (due to the phenomenon of cross-registration of the same trial in different registries, especially for non-North American trials).

## Conclusions

Our study provides cross-sectional data on seamless early-phase oncology trials registered on the ClinicalTrials.gov registry. We highlighted the challenges posed by the evolving landscape of clinical trial design and the issue of missing results within the context of seamless trial design. With a median follow-up of 518 days, only 34.7% trials reported results on Clinical-Trials.gov. The overall reporting rate for Phase 1/2 studies was more than three times higher than for seamless Phase 1 studies. Efforts should be made to adapt the functionality of the Clin-icalTrials.gov database to emerging clinical trial models. One of the overarching issues to be addressed in implementing a seamless design is effectively capturing the nature of staging and sequential cohorts, both in describing trial characteristics and in presenting results.

## Supporting information

**S1 Fig. Flow diagram for identification of seamless early-phase oncology trials.**
(DOCX)

**S2 Fig. Kaplan-Meier curve for the cumulative probability of not reporting clinical trial results over time—Comparison between seamless Phase 1 and Phase ½.**
(DOCX)

**S1 File. Data extraction form.**
(XLSX)

**S1 Table. STROBE statement—Checklist of items that should be included in reports of cross-sectional studies.**
(DOCX)

**S2 Table. ClinicalTrials.gov search parameters.**
(DOCX)

**S3 Table. Details on data categorization for trial characteristics.**
(DOCX)

**S4 Table. Characteristics of a seamless Phase 1 study and a Phase 1/2 study.**
(DOCX)

**S5 Table. Type of cancer.**
(DOCX)

**S6 Table. Details on studies with an expansion cohort directly mentioned.**
(DOCX)

**S7 Table. Characteristics of the endpoints.**
(DOCX)

**S8 Table. Comparison of time to reporting results based on trial characteristics.**
(DOCX)

## Author Contributions

**Conceptualization:** Katarzyna Klas, Karolina Strzebonska, Marcin Waligora.

**Data curation:** Katarzyna Klas.

**Formal analysis:** Katarzyna Klas, Maciej Polak.

**Funding acquisition:** Marcin Waligora.

**Investigation:** Katarzyna Klas, Karolina Strzebonska, Paola Buedo, Alicja Włodarczyk, Samuel Gordon, Paulina Kaszuba.

**Methodology:** Katarzyna Klas, Karolina Strzebonska, Maciej Polak, Marcin Waligora.

**Project administration:** Katarzyna Klas, Marcin Waligora.

**Resources:** Katarzyna Klas.

**Software:** Katarzyna Klas, Maciej Polak.

**Supervision:** Marcin Waligora.

**Validation:** Katarzyna Klas, Karolina Strzebonska, Marcin Waligora.

**Visualization:** Katarzyna Klas, Maciej Polak.

**Writing – original draft:** Katarzyna Klas.

**Writing – review & editing:** Katarzyna Klas, Karolina Strzebonska, Paola Buedo, Alicja Włodarczyk, Samuel Gordon, Paulina Kaszuba, Maciej Polak, Marcin Waligora.

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
