## [Decision Letter · Decision Letter 0]

16 Sep 2024

PONE-D-24-18334Seamless Trials in Oncology: A Cross-Sectional Analysis of Characteristics and ReportingPLOS ONE

Dear Dr. Waligora,

Thank you for submitting your manuscript to PLOS ONE. After careful consideration, we feel that it has merit but does not fully meet PLOS ONE’s publication criteria as it currently stands. Therefore, we invite you to submit a revised version of the manuscript that addresses the points raised during the review process.

I have reviewed the manuscript and you can see my comments in below.

We look forward to receiving your revised manuscript.

Kind regards,

Omid Beiki, M.D., Ph.D.

Academic Editor

PLOS ONE

Journal Requirements:

Reviewers' comments:

Omid Beiki

**Introduction:**

Have you found any previous study on seamless trials in other fields than oncology? Please indicate it in introduction.

Lines 50-53: You have provided 9 references for one sentence. While they seem relevant to the sentence you have written, I prefer you to limit them to only the most relevant ones to the topic of study. Please consider those that you can later use in your discussion section.

**Methods:**

What was the age definition for adults and pediatrics?

Could you please justify why you have selected 24 months for your analyses? Why not 12 months or 36 months? Considering the time to report results provided in table SB, I prefer to use 30 or 36 months to cover at least 75% of trials.

**Results:**

Table 1: I am not sure why you have reported percentages of second column as the ratio to all trials not trials with results. Please either justify the reason or provide the percentage to trails with results for each characteristic. As it is a descriptive table, I prefer to have the percentage to trails with results for each characteristic.

Table 2: It seems you calculated percentage based on all trials. If so, please specify it. Please specify the statistical test and the comparison group for P-value as well.

Table 3: Please specify all variables that were included in the model in table footnote. You should also specify CI and OR in table footnote same as US.

Reviewer's Responses to Questions

**Comments to the Author**

1. Is the manuscript technically sound, and do the data support the conclusions?

Reviewer #1: Yes

2. Has the statistical analysis been performed appropriately and rigorously? 

Reviewer #1: Yes

3. Have the authors made all data underlying the findings in their manuscript fully available?

Reviewer #1: Yes

4. Is the manuscript presented in an intelligible fashion and written in standard English?

Reviewer #1: Yes

5. Review Comments to the Author

Reviewer #1: Abstract:

In the Methods section, please correct "completed between" to "primary completion date" (PCD) to avoid confusion, as there is an important distinction between these terms. Additionally, when describing trials in the registry, it would be clearer to specify "reported as completed (PCD)" since trial status can vary across different registries. This clarification ensures that readers understand the definition being used.

Introduction:

I would describe ClinicalTrials.gov as a relational database rather than just a tool. This distinction is critical for understanding the database's functionalities, such as querying and data extraction. However, consider this comment low priority.

Search Strategy, Extraction form and Reproducibility:

For reproducibility in generating the sample, please provide the exact search string used to identify relevant cancer trials, as mentioned in your protocol. It's important to note that using the term "cancer" may exclude some trials that are still relevant but coded under "neoplasm." While ClinicalTrials.gov searches for synonyms using MeSH terms, certain cancer trials could still be missed. It would be better to attach the extraction form with the variables used. You can include it as a supplementary file, making it accessible as a downloadable file in the supplementary materials.

Table 1 Characteristics:

For better readability, the characteristics in Table 1 should be arranged in descending order of percentages. Additionally, please mention the inter-rater reliability (IRR) in the assessment of the extracted data. Consider this comment low priority.

Cross-Sectional Study Insights:

In cross-sectional studies like this one, the extraction form and the data extracted can be crucial. The study's descriptive analysis should consider the representativeness of the sample, noting that a large proportion of the trials were conducted in North America. This regional concentration may limit the generalizability of the findings, so the results should be interpreted with caution. While the study discusses the results of seamless cancer clinical trials, it would have been more insightful to highlight how seamless trials differ from non-seamless trials in result reporting and characteristics.

Overall Assessment:

Thank you for the opportunity to review this article. My experience in evaluating data from trial registries and developing tools for assessing research practices in biomedical research gives me an appreciation for the importance of this study. However, I must emphasize, as the authors did, the limitations inherent in relying on registry data, particularly regarding trial registration and timely updates. Additionally, the phenomenon of cross-registration or registering the same trial in different registries, especially for non-North American trials, could mean that results might be reported elsewhere, potentially skewing the numbers. Given the North American bias in the sample, there is a regional limitation that should be acknowledged. Since no direct comparison between seamless and other trial designs was made within the sample, the study's primary value lies in its search strategy and extraction form. The meticulous approach to data extraction is commendable, and the authors' efforts to highlight the importance of presenting results for seamless trials, taking into account staging and cohort considerations, is noted.

6. PLOS authors have the option to publish the peer review history of their article (what does this mean?). If published, this will include your full peer review and any attached files.

Reviewer #1: No

---

## [Author Response · Author response to Decision Letter 0]

25 Sep 2024

Dear Editor, Dear Reviewer,

We are grateful for the careful and constructive assessment of our manuscript. We appreciate the opportunity to revise and resubmit it. Please find our responses to the comments in the "Response to Reviewers" file.

Sincerely, 

Prof. Marcin Waligora

Jagiellonian University Medical College

Krakow, Poland

---

## [Decision Letter · Decision Letter 1]

15 Oct 2024

Seamless Trials in Oncology: A Cross-Sectional Analysis of Characteristics and Reporting

PONE-D-24-18334R1

Dear Dr. Waligora,

We’re pleased to inform you that your manuscript has been judged scientifically suitable for publication and will be formally accepted for publication once it meets all outstanding technical requirements.

Kind regards,

Omid Beiki, M.D., Ph.D.

Academic Editor

PLOS ONE

Additional Editor Comments (optional):

Reviewers' comments:

Reviewer's Responses to Questions

**Comments to the Author**

1. If the authors have adequately addressed your comments raised in a previous round of review and you feel that this manuscript is now acceptable for publication, you may indicate that here to bypass the “Comments to the Author” section, enter your conflict of interest statement in the “Confidential to Editor” section, and submit your "Accept" recommendation.

Reviewer #1: All comments have been addressed

2. Is the manuscript technically sound, and do the data support the conclusions?

Reviewer #1: Yes

3. Has the statistical analysis been performed appropriately and rigorously? 

Reviewer #1: Yes

4. Have the authors made all data underlying the findings in their manuscript fully available?

Reviewer #1: Yes

5. Is the manuscript presented in an intelligible fashion and written in standard English?

Reviewer #1: Yes

6. Review Comments to the Author

Reviewer #1: All my comments regarding cross-registration, generalizability of findings, and presentation of results have been addressed. I wish all the best to the authors.

7. PLOS authors have the option to publish the peer review history of their article (what does this mean?). If published, this will include your full peer review and any attached files.

Reviewer #1: No

---

## [Editor Report · Acceptance letter]

20 Nov 2024

PONE-D-24-18334R1 

PLOS ONE

Dear Dr. Waligora, 

I'm pleased to inform you that your manuscript has been deemed suitable for publication in PLOS ONE. Congratulations! Your manuscript is now being handed over to our production team.

Kind regards, 

on behalf of

Dr. Omid Beiki 

Academic Editor

PLOS ONE